# FDI and Firm Productivity: A Comprehensive Review of Macroeconomic and Microeconomic Models

Eleonora Santos 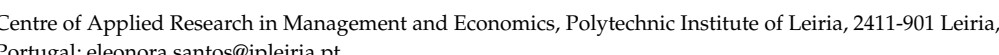

Centre of Applied Research in Management and Economics, Polytechnic Institute of Leiria, 2411-901 Leiria, Portugal; eleonora.santos@ipleiria.pt

**Abstract:** This paper reviews the literature on foreign direct investment (FDI), productivity, and technology upgrading, with a focus on macroeconomic and microeconomic models. It compares the performance of various models used to study FDI and its effects on firms' productivity, via skill and technology upgrading, offshoring, institutional quality, and other related factors. This review highlights the differences and similarities between macroeconomic and microeconomic models, their empirical strategies, and their ability to provide a comprehensive understanding of the mechanisms through which FDI affects productivity and other variables. The empirical literature on the impact of FDI on the productivity of local firms is derived from association studies, which use a neoclassic production function and an augmented Solow-type equation. These models have been shown to be inadequate in capturing the dynamic and complex nature of FDI and the associated externalities, particularly vertical externalities. This paper identifies three criticisms of the literature on pecuniary externalities, including a lack of models that focus specifically on the mechanism of forward linkages, inadequate measures to assess linkage effects, and the failure to include crucial determinant factors in empirical models. Overall, this paper calls for more comprehensive and nuanced models that incorporate the dynamic and complex nature of FDI and its externalities.

**Keywords:** foreign direct investment; productivity; technology upgrading; externalities

## 1. Introduction

Foreign direct investment (FDI) has been a major driver of economic growth in many countries (Mendali 2021). Its positive impact on recipient economies has often been through the transfer of technology (De Mello 1997), knowledge, and skills. One of the key channels through which FDI can be expected to have a positive impact on economic growth is by enhancing productivity (Kenh 2023).The FDI–productivity nexus has been the subject of many empirical studies (e.g., Zhao and Zhang 2010; Huynh et al. 2021; Malikane and Chitambara 2017; Deng et al. 2008) aimed at examining the existence and strength of this relationship. Some studies have found that FDI can lead to productivity externalities, while others have found little or no evidence of such externalities. Despite these findings, there is still no consensus on the exact nature of the relationship between FDI and productivity (Lim 2001; Marasco et al. 2023). This paper provides a comprehensive review of the models addressing the FDI–productivity nexus on the host and country of origin of MNCs to identify the gaps in the literature. It begins by reviewing the mechanisms through which FDI can affect productivity, with a particular emphasis on externalities and technology transfer. The theoretical foundations of the nexus are examined by analysing how different scholars have conceptualised the relationship between FDI and productivity in economic models. It then turns to the empirical literature, discussing the typical method used to estimate the relationship between FDI and productivity. This review highlights that while there is a substantial body of literature examining the FDI–productivity nexus, there is no consensus on the approach used to investigate the relationship. Different studies have focused on different aspects of the relationship, and this has led to a broad range of

results. Overall, this review underscores the importance of FDI as a driver of productivity and economic growth and the need for models that can capture the complexity of the transmission channels through which productivity externalities are transmitted.

Insofar as it evaluates the adequacy of theoretical models in providing empirical evidence of productivity externalities via the various transmission mechanisms prompted by FDI, this paper contributes to the existing literature on MNCs. It presents a critical analysis of the current state of research on FDI and highlights the limitations of existing FDI–productivity models. By doing so, this article sheds light on the complexities of FDI and provides insights for policymakers and researchers to better understand how FDI can contribute to economic development. Overall, this article makes a valuable contribution to the existing literature on FDI models by offering a fresh perspective on the topic. Future research should focus on identifying the conditions under which the FDI–productivity nexus is strongest.

Governments can use the results of this study to design policies that attract FDI to their countries and promote productivity growth. In particular, policies that are targeted towards attracting high-quality FDI that can generate positive externalities can be more effective in promoting long-term growth.

The remainder of this article is structured as follows. Section 2 provides an examination of technological and pecuniary externalities; Section 3 focuses on models of pecuniary externalities, specifically discussing macroeconomic models and macroeconomic models with microeconomic foundations; Section 4 analyses models of FDI–productivity that incorporate heterogeneous firms and workers; Section 5 evaluates models of offshoring and productivity; Section 6 explores models of institutional quality–FDI–productivity; Section 7 presents the empirical literature; Section 8 engages in a discussion; and finally, Section 9 concludes, summarizing key findings, policy implications, and future directions for research.

## 2. Technological and Pecuniary Externalities

Externalities occur when the entrance of an MNC leads to increases in the productivity of local firms and when MNCs are not able to internalise the full value of these benefits (Bencivelli and Pisicoli 2021). Technological externalities occur when the geographical proximity facilitates the dissemination of techniques and processes from MNCs to local firms without costs. Pecuniary externalities take place when the contacts between MNCs and local firms prompt the latter firms to use specialised inputs that cause reductions in costs or increases in revenues, and, as a result, create an increase in productivity. In contrast to technological externalities, pecuniary externalities do not affect the production function of the firm; instead, they impact the profit function.

According to Marshall (1890a), economies of scale arise via three external sources: the effects of intersectoral coordination, i.e., the advantages associated with pecuniary externalities arising from sharing inputs with all firms; the effects of interfirm knowledge externalities, i.e., the advantages associated with technological externalities; and gains from labour pools i.e., the advantages associated with both technological and pecuniary externalities. Thus, pecuniary externalities are related to nontechnological innovations, rather than technological innovations that occur via technological externalities. This distinction between technological and nontechnological innovations was introduced by the Organization for Economic Cooperation and Development (2005) and by the Eurostat through the Oslo Manual. The first includes product and process innovations while the latter includes marketing and organisational innovations. However, marketing innovations often coincide with product innovations, while firms with organisational innovations often introduce new technological processes (Bunduchi and Candi 2021).

In developed countries, nontechnological innovations occur in a significant share of manufacturing firms (Henao-García and Cardona Montoya 2023). For example, according to the Community Innovation Survey CIS—2006, in 2004–2006, on average 45% of the manufacturing firms from 19 countries were nontechnological innovators. Additionally,

according to the Eurostat, Germany, Luxembourg, and Portugal are the latest top marketing innovators. By providing information (technical, economic, and organisational) to their suppliers and supplying better quality inputs to their customers (producers of final goods) MNCs allow local firms to reduce their production costs. The benefits to upstream firms arise when the increased demand from MNCs allow for economies of scale that reduce the production costs and increase productivity, since the same number of sales now incorporates cheaper inputs. This demand effect is crucial in the models of backward linkages. Along the same lines of the literature on technological change, the benefits to customers (producers of final goods) arise from the intensive use of better quality/cheaper inputs that are likely to increase their productivity (Deng et al. 2008).

Productivity externalities consist of increases in the productivity of local firms because of foreign presence in the host economy (Bruhn et al. 2020). Thus, productivity externalities arise not only from technological externalities but also from pecuniary externalities. In fact, from my point of view, productivity externalities are a result of both technological and pecuniary externalities often working together. Indeed, technological externalities give rise to pecuniary externalities, and vice versa. For example, consider that FDI leads to an innovation of the production process of local suppliers that favours capital over labour. This may lead to a change in the organisational structure with fewer horizontal or vertical levels. In this case, technological externalities cause pecuniary externalities due to cost decreases via the reduction of the number of workers. In other words, technological innovations caused nontechnological innovations. Another example is what Antonelli and Feder (2021) call pecuniary knowledge externalities (PKE), i.e., technological externalities caused by pecuniary externalities. According to Durongkaveroj (2021), the effect of demand in downstream sectors favours increased levels of division of labour in upstream sectors. The division of labour raises the levels of expertise and may cause pecuniary externalities, such as lower prices for capital goods and intermediate inputs.

Moreover, considering that an important part of R&D aims to identify and develop idiosyncratic resources that can be used extensively by firms to reduce the production cost, then the abundant use of cheaper inputs in upstream sectors reduces the cost of R&D and, thus, increases the TFP in the downstream sector. This virtuous cycle may explain why small firms with low levels of R&D are able to introduce rapid rates of innovation (Peters et al. 2021). Hence, the externalities mechanisms are complex and interdependent. Sometimes it is hard to distinguish the effects of both types of externalities on the productivity of local firms. Therefore, the theoretical literature deals with many difficulties and gaps and, so far, it has not provided sound predictions for empirical research. Moreover, despite the importance of pecuniary externalities and nontechnological innovations to the productivity of local firms, researchers have only recently begun to incorporate them into models in a context of linkages suppliers and customers under imperfect competition and increasing returns (e.g., Markusen and Venables 1999). One of the reasons is perhaps the difficulty in assigning a specific type of externality to a particular transmission mechanism since each transmission channel may originate from more than one type of externality. As a result, up until now, there have been no attempts to incorporate vertical externalities in growth models and the few attempts that do include horizontal externalities in such models are in a North–South (representing the most and the least developed countries, respectively) product-cycle framework.

## 3. Models of Pecuniary Externalities

By and large, models of vertical externalities use a common set of tools involving the combination of monopolistic competition as in Dixit and Stiglitz (1977) and 'iceberg' trade costs to deal with increasing returns due to specialisation, trade costs, and input–output (IO) linkages. In addition, according to Marshall (1890b), external economies of scale are required to ensure that firm productivity increases with the increasing size of the economy. Such models typically analyse the economy from the demand side. Therefore, a fundamental assumption is that inputs are not tradable on the international markets,

so that MNCs are forced to buy their inputs locally. Another implicit assumption is that the MNC has a superior technology to allow for productivity gains for local firms via linkages. The key mechanism of these demand-side models is the following: The entry of the MNC in the local market increases the demand for intermediate goods (or services), therefore increasing the number of producers of such goods and, since the market operates in monopolistic competition, increasing its variety. Since, like in Ethier (2014), these models assume love for the variety of inputs in the production of the final good; the increased variety of intermediate goods raises the productivity of the final good.

*3.1. Macroeconomic Models*

These are endogenous growth models with product innovation via increasing the diversity of intermediate inputs following the works of Grossman and Helpman (1991b) and Romer (1990). In a general equilibrium framework, these authors explain the long-run growth by explicitly modelling the innovation behaviour of firms in the context of monopolistic competition and increasing returns to scale. Technological progress assumes the form of horizontal product differentiation (capital goods for Grossman and Helpman (1991b) and consumer goods for Romer (1990)). This allows for static gains from access to a wide range of products as well as dynamic gains through an increase in the rate at which the new varieties are introduced as in Ethier (1982). Ultimately, the constant improvement of the intermediate inputs causes increases in the productivity of the final goods, and consequently, economic growth.

The Grossman and Helpman (GH) model (1991b) is a widely used endogenous growth model that incorporates product innovation through the increasing diversity of intermediate inputs. It departs from a production function as follows:

$$Y = F(K, L, G) \tag{1}$$

where Y is the aggregate output, K is the capital, L is the labour, and G represents the set of intermediate goods available for production.

The production function F is assumed to exhibit constant returns to scale, which implies that output per capita is constant in the long run. The function G represents the set of final goods produced using intermediate goods. The equilibrium condition of the GH model is the balance between investment in physical and innovation capital, which is expressed as

$$I(t) = \delta K(t) + cG(t) \tag{2}$$

where I(t) is the investment in physical and innovation capital, $\delta$ is the depreciation rate, and c is the rate of investment in innovation capital.

The Romer (1990) model is a variant of the GH model that focuses on product innovation as the main driver of endogenous growth through increasing returns to scale at the firm level. It can be represented as follows:

$$Y = AK^\alpha(LH)^{(1-\alpha)} \tag{3}$$

where Y is the output, K is the capital, L is the labour, A is the technology, H is an index of the variety of intermediate goods, and $\alpha$ is the share of capital income in the output.

The production function exhibits increasing returns to scale due to the variety of intermediate goods. The equilibrium condition of the Romer model is expressed as

$$rI(t) = \delta K(t) + cH(t) \tag{4}$$

where r is the user cost of capital and H(t) represents the investment in the variety of intermediate inputs.

The models of Rivera-Batiz and Rivera-Batiz (1990), Rodriguez-Clare (1996), and Alfaro et al. (2004) are based on the works of Ethier (1982), Helpman (1984), Markusen

(1984), Romer (1990), Grossman and Helpman (1991a), Rivera-Batiz and Romer (1991a, 1991b) and Romer (1986).

The models of Rivera-Batiz and Rivera-Batiz (1990) and of Rodriguez-Clare (1996) are particular cases of the model of Alfaro et al. (2004) to the extent that they use the same specification and hypotheses, but they are not as complete regarding the number of inputs. Indeed, Rivera-Batiz and Rivera-Batiz (1990) only considers the physical capital and Rodriguez-Clare (1996) only considers the labour and Rivera-Batiz and Rivera-Batiz (1990) carries out a static analysis.

The model of Rivera-Batiz and Rivera-Batiz (1990) explores the impacts of FDI in the context of increasing returns due to specialisation. It examines a small open economy that produces two tradable goods: good X and good Y. The economy consists of firms that produce these goods, households that supply labour to the firms, and a government that collects taxes and provides public goods. Firms are assumed to face increasing returns due to specialisation. The production function for good X is given as

$$X = A \, (\theta(EL))\mu \, (1 - \theta(EX))(1 - \mu) \tag{5}$$

where A refers to total factor productivity, EL refers to domestic labour, EX refers to foreign labour, $\theta$ is a measure of the degree of specialisation of production, and $\mu$ is the elasticity of substitution between domestic and foreign labour.

The production function for good Y is similar to that for good X. The equilibrium condition of the model can be described as

$$P = ((1 - \tau) + \tau(e/(1 + e))) \, (w/A) \tag{6}$$

where P is the relative price of good X to good Y, $\tau$ is the tariff rate, e is the degree of nominal exchange rate flexibility, w is the nominal wage rate, and A is the total factor productivity.

The equilibrium condition shows that the relative price of good X and Y is affected by tariff rates, nominal exchange rate fluctuations, and nominal wage rates. This equation is used to analyse how the presence of FDI affects output, specialisation, and welfare in the domestic economy.

The Rodriguez-Clare model (1996) introduces the concept of forward linkages, which describe the inputs to the foreign affiliate by local suppliers. The production function is

$$Y = F(K, L, Z) \tag{7}$$

where Y is the total output, K is the capital stock, L is the labour force, and Z is the vector of intermediate inputs.

The output function F is the Cobb–Douglas production function and is given as

$$F(K, L, Z) = A(K\alpha L1 - \alpha) \times \prod (zi1 - \rho(i))\rho/ \, 1 - \rho \tag{8}$$

where A is total factor productivity, $\alpha$ is the share of capital in the production process, $\rho$ is the degree of substitution between intermediate inputs, and zi is the level of intermediate input i.

The central idea behind the Alfaro et al. (2004) model is that FDI inflows can stimulate economic growth by improving local financial markets. The model departs from

$$G = A(K, L, S) \tag{9}$$

where G is the rate of economic growth, K is the capital stock, L is the labour force, and S is the availability of local financial institutions.

The production function A is given as

$$A(K, L, S) = A* \, (K/L)\hat{\ }\alpha \times S\hat{\ }\beta \tag{10}$$

where A* is the level of total factor productivity, $\alpha$ is the share of capital in the production process, and $\beta$ is the share of financial institutions in the production process.

Overall, the models of Rivera-Batiz and Rivera-Batiz (1990), Rodriguez-Clare (1996), and Alfaro et al. (2004) explore different aspects of the relationship between FDI and economic development, highlighting the importance of factors such as specialisation, knowledge spill overs, and local financial markets in determining the impact of FDI on growth. All these models contribute to our understanding of the complex interplay between FDI, institutional quality, and economic growth.

## 3.2. Macroeconomic Models with Microeconomic Foundations

Partial equilibrium models of vertical FDI externalities are macroeconomic models with microeconomic foundations. Like in the previous models, there are two sectors, the final good and the intermediate good, and the mechanism for backward linkages is the same as described above. However, now, the presence of MNCs has two opposite effects. Besides the positive demand effect, there is the negative effect of the competition. Indeed, the increase in total output due to the presence of MNCs may cause a decrease in the market price that forces the exit of some less efficient local firms. This category of models includes Markusen and Venables (1999) and Lin and Saggi (2004).

The model of Markusen and Venables (1999) performs an analysis on the demand side where the market for intermediate goods operates in monopolistic competition, while Lin and Saggi (2004) performed an analysis on the supply side where the market for intermediate goods operates in a Cournot-fashion oligopoly. The choice of this latter market structure is required to analyse the strategic behaviour between MNCs and local firms. Indeed, strategic interfirm rivalry is likely to exert a major influence on the equilibrium outcomes.

To demonstrate the two effects described, both models assume that the demand for the final good depends on the requirements of the intermediate goods per unit of final good and on the price of the intermediate good.

In their model, Markusen and Venables (1999) explored the possibility of FDI acting as a catalyst for industrial development. The production function is

$$Y = f(K, L) \tag{11}$$

where Y is output, K is the capital input, and L is the labour input.

The capital input is subdivided into two parts: KRD, or region-specific physical capital, and KURD, or region-specific human capital.

$$K = KRD + KURD \tag{12}$$

The local R&D sector is given as

$$RD = aRD \, (Y/RD)^{\wedge}(1/\theta) \tag{13}$$

where RD is the amount of R&D investment, ard is the productivity of the R&D sector, Y/RD is the output per R&D dollar, and $\theta$ is the elasticity of R&D with respect to the output.

The investment in a region depends directly on local R&D; however, this relationship is subject to the influence of global R&D input:

$$I = I^*(RDf/RD)\theta \tag{14}$$

The global R&D input, which is the sum of the R&D activities of other regions, is weighted by the inverse of the distance between regions (Dij) raised to the power of $\alpha$.

$$RDf = \Sigma j \, RDj/(Dij)^{\wedge}\alpha \tag{15}$$

The share of FDI in investment in a region (Ri) is given as

$$Ri = Gi \times (FDI/R) + (1 - Gi) \times (UFPi/RPi) \tag{16}$$

which depends on the quality of governance or institutional quality (Gi) and the competitiveness of the region in terms of unit factor costs (UFPi) relative to the corresponding units in the home country (RPi). R is the total investment in a region. Equilibrium is established when all markets are in balance, i.e., total supply equals total demand.

Lin and Saggi (2004) developed a theoretical framework to examine the impact of ownership structure on the choice of technology and incentives to invest in technology upgrades in international joint ventures. The model assumes the production function:

$$Yi = (1 - \alpha i) \times ((li)^{\wedge}(-bi)/(-bi + 1)) \times (Ti \times H)^{\wedge}bi \tag{17}$$

where Yi is output, li is labour input, Ti is the level of technology employed, H is the level of human capital, $\alpha i$ is the share of ownership held by the foreign partner, and bi is the elasticity of labour with respect to output. The unit costs of production are given as

$$Ui = Ui (Ti, Hi, \alpha i, \theta i) \tag{18}$$

where $\theta i$ is the cost of technology transfer and Ui is the share of labour costs in total costs.

The choice of technology depends on the level of human capital, the share of ownership held by the foreign partner, and the level of technology employed in the industry in the home country (Fi).

$$\Delta Ti = \lambda i \times (\alpha i Hi + (1 - \alpha i) Fi) \tag{19}$$

where $\Delta Ti$ represents the change in technology from one period to another.

Total profit (Πi) is the difference between total revenue (Ri × yi) and total costs (Ui × Ri), where Ri represents the rate of return on foreign investment:

$$\Pi i = (1 - \alpha i) \times Ri \times yi \tag{20}$$

Equilibrium is established when all markets are in balance, i.e., total supply equals total demand.

To sum-up, while Markusen and Venables (1999) focused on the impact of FDI on the level of investment, production, and overall economic growth, with a specific emphasis on the role of global R&D flows and institutional quality, Lin and Saggi (2004) analysed the impact of ownership structure on technology choices and incentives to invest in technology upgrading in joint ventures. Both models emphasize the role of incentives and market mechanisms as a driving force in the decision-making process while considering the specific institutional and economic contexts of the regions/joint ventures under consideration.

## 4. Models of FDI–Productivity with Heterogeneous Firms and Workers

FDI can lead to technology upgrading through knowledge externalities and the introduction of new technologies (Fu and Diez 2010; Wang et al. 2020; Ning et al. 2023; Grossman and Hart 1986). Acquiring new knowledge and technology can be particularly important for less productive firms in order to catch up with more productive ones.

FDI can also lead to skill upgrading through the transfer of knowledge and technology, as well as the introduction of new management practices (Melitz 2003; Melitz and Ottaviano 2008; Harrison and McMillan 2010; Arkolakis et al. 2012). This can lead to higher wages and productivity gains for both skilled and unskilled workers. For example, in their study on Mexico, Harrison and McMillan (2010) found that FDI leads to higher wages and productivity gains for skilled workers, but also benefits unskilled workers through an externality effect.

In their seminal paper, Melitz and Ottaviano (2008) showed that FDI can lead to productivity gains both for foreign affiliates and domestic firms through knowledge externalities and skill upgrading.

This model is a heterogeneous-firm model that incorporates firm-level productivity differences and trade costs. It assumes that: (1) There are a large number of heterogeneous firms in each country and these firms have to decide whether to enter the export market or remain in the domestic market. (2) There are two countries, denoted by i and j, and each country has a continuum of firms indexed by the parameter "z". Each firm has a productivity level denoted by "$\phi$". The productivity levels are distributed according to a known probability density function "$G(\phi)$". (3) Firms face a fixed cost of exporting, denoted by "F", and a variable cost of exporting, denoted by "c". The variable cost of exporting is assumed to be proportional to the distance between the two countries, denoted by "d".

The profits of a firm that chooses to export are provided by the following equation:

$$\pi_i(z) = \phi_i(z) - c\, d_{ij} - F \tag{21}$$

where $\pi_i(z)$ is the profit of a firm in country i with productivity level $\phi_i(z)$ that exports to country j, and $d_{ij}$ is the distance between the two countries.

The equilibrium condition in the model is that all firms that choose to export must earn positive profits. This implies that the variable cost of exporting must be less than the revenue gained from exporting, given as

$$p_j \geq c\, d_{ij} + F/\phi_i \tag{22}$$

where $p_j$ is the price of the firm's product in country j, and the inequality holds for all firms in country i that choose to export.

The equilibrium condition determines the set of firms that export and their productivity levels. The model shows that firms with higher productivity levels are more likely to export and earn higher profits and that trade liberalisation can lead to gains in productivity through increased competition and the reallocation of resources to more productive firms.

## 5. Models of Offshoring and Productivity

Offshoring can also have productivity effects through the reallocation of tasks and the increase in competition (Antràs and Helpman 2004; Antràs et al. 2006; Helpman 2006; Markusen and Venables 2007; Goldberg et al. 2010; Hijzen et al. 2011; Bernard et al. 2011; Baldwin 2012). By reallocating low-skilled tasks to low-wage countries, firms can focus on their core activities and increase their productivity. Offshoring can also increase competition and exert pressure on firms to become more productive. In their study on Sweden, Hijzen et al. (2011) found that offshoring leads to productivity gains for firms that are more productive to begin with but also to productivity losses for less productive firms.

The Hijzen et al. (2011) model assumes that firms have a production function that depends on labour and capital inputs, denoted by "L" and "K", and the level of outsourcing, denoted by "O". The production function can be expressed as follows:

$$Y = A(L, K, O) \tag{23}$$

where Y is the output, and A is a function that captures the productivity of the firm.

The level of outsourcing, "O", is assumed to be endogenous and determined by the firm's decision to outsource. The decision to outsource is assumed to be based on the cost of outsourcing, denoted by "C", and the productivity gains from outsourcing, denoted by "G". The cost of outsourcing is assumed to be a function of the wage differential between the home country and the outsourcing destination, denoted by "$w_{f}$" and "$w_{o}$", respectively, as well as the cost of communication, denoted by "c".

The productivity gains from outsourcing are assumed to depend on the quality of inputs, denoted by "q", and the firm's ability to manage the outsourcing process, denoted by "M".

The firm's decision to outsource can be expressed as follows:

$$O = h(C, G) \tag{24}$$

where h is a function that captures the firm's outsourcing decision. The equilibrium condition can be expressed as

$$G > C \tag{25}$$

where G is the productivity gains from outsourcing, and C is the cost of outsourcing. Firms will outsource if the productivity gains from outsourcing exceed the cost of outsourcing.

The model predicts that firms with higher productivity levels are more likely to outsource and that outsourcing can lead to productivity gains through access to higher-quality inputs and the ability to leverage the outsourcing destination's comparative advantage. The model also suggests that policy interventions that reduce the cost of communication or improve the quality of inputs can lead to increased outsourcing and productivity gains.

## 6. Models of Institutional Quality–FDI–Productivity

According to research by Jung (2020) and Jimenez et al. (2021), there is a strong relationship between institutional quality, foreign direct investment (FDI), and productivity.

Several theoretical models have been proposed to explain this relationship: (1) The quality of institutions model, which proposes that institutional quality is a key determinant of the level of FDI inflows into a country. Higher quality institutions, such as those that are transparent, accountable, and stable, help to create a more conducive environment for foreign investors. This, in turn, leads to higher levels of FDI, which can boost productivity through the transfer of technology and know-how. (2) The complementarity model, which suggests that institutional quality and FDI can complement each other in driving productivity growth. In this model, higher levels of FDI can lead to improvements in institutional quality by promoting competition and demanding better regulatory environments. In turn, better institutions can attract even more FDIs, creating a virtuous cycle of investment and productivity growth. (3) The institutional coherence model, which argues that institutional quality is a necessary but insufficient condition for effective use of FDI in boosting productivity. In this model, the effectiveness of FDI in driving productivity growth depends on the coherence and coordination between different institutions. This includes macroeconomic policies, investment policies, and a favourable business environment, among others.

These models typically take the form of a production function:

$$Q = F(K, L, A, I) \tag{26}$$

where Q is the output (often measured as GDP or productivity), K is the physical capital, L is the labour, A is the technology (or other exogenous variable), and I is the institutional quality.

In this production function, institutional quality acts as a "complement" to the other inputs, meaning that it can boost output beyond what is possible with only physical capital, labour, and technology. Therefore, the contribution of institutional quality to output is positive and significant in most cases.

To derive the equation of equilibrium, we can begin with a basic economic model where the foreign investor seeks a profit-maximizing return on investment (ROI) in a host country with a given institutional quality index. This can be represented by the following equation:

$$ROI = F(K, L, A, I) - rK - wL \tag{27}$$

where ROI is the return on investment, F is the production function, r is the rental cost of capital, and w is the wage rate for labour.

In this equation, the investor will allocate capital and labour based on their marginal productivity and the relative cost of each factor in the host country.

The equilibrium occurs when the expected ROI is equal to the required ROI, represented by the following equation:

$$ROI^* = ROIreq \tag{28}$$

where ROI* is the expected ROI, and ROIreq is the required ROI for the investor to justify their investment in the host country.

Solving for equilibrium requires determining the values of K and L that satisfy this condition, given the exogenous variables A and I. This can be done using numerical methods or by applying differential calculus to the production function.

The equilibrium value of I can then be obtained as the optimal level of institutional quality that maximizes ROI and ensures the investor's investment will be profitable.

Overall, these models show that institutional quality plays a critical role in attracting FDI and boosting productivity and that equilibrium can be achieved by optimizing the mix of physical capital, labour, and institutional quality in the host country. By addressing institutional weaknesses and creating a more stable and coherent environment for foreign investment, countries can harness the potential of FDI to drive long-term economic growth.

## 7. Empirical Literature

Existing empirical studies exploring the relationship between FDI and productivity have shown that downstream effects of FDI are generally more beneficial than upstream effects (Santos et al. 2023). In the context of forward linkages, MNCs transfer knowledge and technology to their suppliers, which helps to enhance the technological capabilities and operational efficiency of local firms. Empirical studies have explored these associations, using neoclassical production functions to estimate the productivity of local firms as a residual, representing a function of externalities from FDI and a set of control variables (Blomström and Kokko 1997; Alfaro et al. 2004; Keller 2004). These studies demonstrate the importance of technical knowledge and the need for the development of backward and forward linkages between local firms and MNCs in driving the productivity of firms upward. However, the empirical debate has been substantially centred around backward linkages (Alfaro et al. 2004), while less attention has been paid to the mechanisms behind forward linkages (Santos and Khan 2019). While these studies do provide valuable insight, there is still a lack of models that specifically consider the mechanism of forward linkages between MNCs and local firms.

The available theoretical models focus more on the externality effects of FDI on the local economy as a whole. Furthermore, theoretical models do not provide adequate measures to assess linkage effects or strong empirical predictions for researchers, and these models do not address critical questions about what determinant factors ought to be included in empirical models illustrating the transmission mechanism of vertical externalities. As such, there is inadequate data on the direct effects of forward linkages, failing to capture their true extent. Furthermore, the measures used to assess the linkage effects of FDI have been entirely proxy-based, such as the composition of local content in MNCs production (Jordaan 2008), which leads to the inadequacy of capturing the true extent of the relationship between FDI and productivity. This is because a high degree of local content does not necessarily translate to increased linkages. The inputs used by local firms may be for low-value-added activities that do not contribute to enhancing technological and operational capabilities.

There is also a lack of inclusion of important determinant factors in empirical models. For example, the size and nature of the industry, the local competencies and capabilities of the firms and MNCs, and the institutional environment in host countries are all important factors that can influence the FDI–productivity nexus through forward linkages (Ning et al. 2023). Neglecting to include these key factors may lead to incomplete and inac-

curate measures and recommendations to policy makers for enhancing relevant policies for host countries.

Another caveat in the literature lies in the fact that the use of production functions is not always appropriate and may result in a lack of understanding of the generation of knowledge externalities. This would require researchers to specify a knowledge-generation function, taking into account that the internal and external knowledge are complementary inputs, with MNCs functioning primarily as knowledge integrators (Antonelli and Feder 2021).

Improvements have been made to the methods used during empirical research, using both statistical analysis and econometric models to study the relationships between FDI, productivity, and other factors. However, these models have their differences, stemming from the data sources, statistical methods used, the specific variables studied, and the underlying mechanisms that drive these relationships. While these models do have similarities, the differences in their assumptions, methods, and areas of interest make comparison difficult. Thus, it is crucial to address these differences to develop a comprehensive understanding of the relationship between FDI and productivity to capture the transmission channels of externalities. In addition to developing a more comprehensive knowledge base, a theoretical understanding of the process of transmitting forward linkages is also necessary.

In summary, although the empirical literature indicates that forward linkages contribute significantly to productivity growth, there is still a need to develop models that focus specifically on these mechanisms and improve upon the measures used to assess the linkage effects. The focus must be directed towards the generation of knowledge externalities, which are complementary inputs instead of production functions. Including crucial determinant factors in empirical models can contribute to better policy recommendations for host countries to attract and benefit from FDI.

## 8. Discussion

Several types of macroeconomic and microeconomic models have been applied to study various aspects of FDI, productivity, and technology upgrading. From the analysis above, we can systematise the following findings: (1) Macroeconomic models are typically designed to analyse and forecast the overall performance of the economy, while macroeconomic models with microeconomic foundations are built on detailed microlevel data and aim to explain the underlying mechanisms and processes that drive macroeconomic trends. They both seek to explain how FDI and other factors impact overall economic performance. However, macroeconomic models with microeconomic foundations are often more detailed and can provide more insights into the specific channels through which FDI affects productivity and other outcomes. (2) Models of FDI and technology upgrading often focus on the role of MNCs in diffusing new technologies to host countries. These models typically assume that MNCs have access to superior technology and that they transfer this technology to domestic firms in the host country. They try to explain how FDI leads to the upgrading of technology and productivity in host countries. However, these models often differ in their assumptions about the types of technologies being transferred, the nature of the relationship between MNCs and local firms, and the impact of FDI on human capital and other factors. (3) Models of FDI and skill upgrading focus on how FDI affects the skill levels of workers in host countries. They typically assume that MNCs bring advanced technologies and management practices to host countries and that exposure to these practices leads to skill upgrading among local workers. They examine the impact of FDI on human capital accumulation, but these models often differ in their predictions about the specific types of skills that are most affected, the channels through which skill upgrading occurs, and the effects of skill upgrading on wider economic outcomes such as productivity, wages, and employment. (4) Models of offshoring and productivity focus on how firms in high-wage countries outsource production to low-wage countries and how this affects their productivity. These models typically assume that firms engage in offshoring to reduce costs and that this leads to productivity gains. They examine how international trade affects productivity, but these models often differ in their assumptions

about the motivations for offshoring, the nature of the production process, and the impact of offshoring on employment, wages, and other factors. (5) Models of institutional quality, FDI, and productivity focus on how a country's institutional environment affects the impact of FDI on productivity. These models typically assume that a better institutional environment encourages more productive forms of FDI, reduces the risk of expropriation, and creates a more favourable business environment. They examine the role of institutions in facilitating productive FDI, but these models often differ in their assumptions about the specific institutional mechanisms that are most important and how they interact with other factors such as human capital and market structure.

Regarding empirical studies, this paper identifies two main limitations. Firstly, there is an issue with the measures used to assess the linkage effects of FDI. Many studies have relied on simple proxies, such as the degree of local content in MNCs' production, which do not adequately capture the true extent of the linkages. Secondly, there is a failure to include crucial determinant factors in empirical models, such as the size and nature of the industry, the local competencies, and capabilities of the firms and the MNCs, the institutional environment in host countries, and the technology transfer mechanisms. Addressing these gaps in the literature is crucial for a more comprehensive understanding of the linkages between FDI and productivity and for better policy recommendations for host countries to attract and benefit from FDI.

## 9. Conclusions

FDI is a driving force in global economic growth and development. It is not only important for the host countries' economies but also for the investing countries. Over the years, there has been extensive literature written on FDI, covering various perspectives ranging from its effects on productivity to its impact on host country institutions. The literature on FDI has grown in both theoretical and empirical contributions, making it challenging to understand its overall impact fully. In this paper, I presented a review of the literature on FDI and its effects on productivity, specifically focusing on the externalities that may arise from FDI, including technology upgrading, skill upgrading, institutional quality, and offshoring. I discussed the theoretical models underpinning the FDI and productivity nexus and then delved into empirical studies that have explored the subject.

This review of the literature on FDI and productivity has highlighted several key findings. First, while theoretical models predict various channels through which FDI can have externality effects on local firms, empirical evidence is mixed and requires careful evaluation. Second, methodological challenges in measuring and quantifying the magnitude of productivity externalities from FDI remain significant. Third, the literature has identified several factors that could influence the nature and magnitude of FDI externalities, including the technology gap between local and foreign firms, absorptive capacity, institutional quality, and human capital. Furthermore, my analysis revealed that most studies have focused on backward linkages and their impact on local suppliers, whereas forward linkages and their implications for local demand have received less attention. Avenues for future research could explore the interplay between backward and forward linkages and assess how different sectors and regions within countries might respond differently to FDI inflows. Overall, this review underscores the importance of deeper and more nuanced approaches to analysing the relationship between FDI and productivity externalities. It is critical to consider the heterogeneity both of FDI inflows and of the local firms that receive them, as well as broader institutional and policy factors that shape the channels through which FDI can generate externalities. Further research that more explicitly incorporates these factors can provide valuable insights into how policy can best promote sustainable and inclusive economic growth through international investment.

**Funding:** This research was financed by the National Funds of the FCT—Portuguese Foundation for Science and Technology within the project UIDB/04928/2020 and under the Scientific Employment Stimulus—Institutional Call CEECINST/00051/2018.

**Institutional Review Board Statement:** Not applicable.

**Informed Consent Statement:** Not applicable.

**Data Availability Statement:** Not applicable.

**Conflicts of Interest:** The author declares no conflict of interest.

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
