# Peer review of "FDI and Firm Productivity: A Comprehensive Review of Macroeconomic and Microeconomic Models"

_economies, doi:10.3390/economies11060164_

Round 1

Reviewer 1 Report (Previous Reviewer 1)

The authors have made an appropriate effort to address my comments and improve the paper.

Minor editing of English language is required.

Reviewer 2 Report (Previous Reviewer 2)

Paper is improved.

This manuscript is a resubmission of an earlier submission. The following is a list of the peer review reports and author responses from that submission.

Round 1

Reviewer 1 Report

This manuscript aims to review the FDI-induced externality literature by focusing on the theoretical and empirical models of vertical externalities, and summarizes that so far, the theoretical insights of the transmission mechanisms of vertical externalities to firms’ productivity are not fully exploited and that, with few exceptions, theoretical models do not provide adequate measures to assess the linkage effects.

Overall, I think that the manuscript reviews appropriately an important part of the literature in a structured and informative way. Even so, I think that it lacks some recent and important literature on this topic and can be improved further with a relatively small investment of time. Below are some comments and suggestions that I would like to make.

  • Current manuscript reviews only traditional theoretical models with homogeneous firms and workers. However, recent strong research direction in this literature is modeling with heterogeneous firms and workers, and explaining theoretically the link between FDI (or more broadly globalization) and productivity. Papers in this literature have highlighted the FDI(offshoring)-induced growth effect through technology and skill upgrading mechanisms. Given that the main purpose of this manuscript is to review the literature, I would suggest to add a section or subsection of this literature with some relevant references.
  • As another important branch of this literature, the institutional quality-FDI-productivity links have also been emphasized in a huge amount of papers. This literature may also be of importance enough to be treated in a separate section or subsection. Anyway, it would be nice to discuss the theory literature with some relevant references. Actually, as a review paper, current manuscript appears to be quite short.
  • With the current title and abstract of the manuscript, the readers may expect some general review of the literature on the total externality effects of FDI in both home and host countries. Current manuscript, however, seems to focus only on the host country effects with inward FDI. The focus should be clearly stated in the text, and at least in the abstract.
  • There is a typo in the section numbering. Maybe the author(s) wanted to organize the theoretical models with subsections?

Reviewer 2 Report

The paper is a review of the existing literature on externalities from FDI, however it only summarizes theoretical models and empirical findings of these literature are not discussed at all. 

Author(s) should better highlight how this article contributes to the strand of the existing literature, and how this article could be useful both in terms of new research results and in terms of methodologies employed in the quantitative analysis.

Reviewer 3 Report

In my opinion, the article does not meet either the substacve or the formal criteria for publication.

There is no correct structuring of the article and no logical line for the reader to follow in order to have a clear picture of the externalities of FDI. Some parts( e.g. 3) of the article are insufficiently treated and do not have clear references.

In almost the entire text of the article, instead of mentioning the models, authors or articles referred to, only the number corresponding to the bibliographic reference is mentioned (eg line 68: „According to [15], economies of scale...”)

Also, the formulas presented are not correctly related (e.g. in line 185, it is mentioned that relation 6 is substituted in relation 4 but the last relation presented is no.3; in line 315, reference is made to the logarithm of relation 1 !!??).

I consider that, in order to publish this paper, it is necessary to redo the research, emphasizing the logical exposition of the demonstration that the authors want to bring to the readers' attention.